



# A reconstruction of the ice thickness of the Antarctic Peninsula Ice Sheet north of 70ºS

Kaian Shahateet[1,2,3], Johannes J. Fürst[2], Francisco Navarro[1], Thorsten Seehaus[2], Daniel Farinotti[4,5], and Matthias Braun[2]

[1]Depto. de Matemática Aplicada a las TIC, ETSI de Telecomunicación, Universidad Politécnica de Madrid, Madrid, Spain
[2]Insitute of Geography, Friedrich-Alexander-Universität Erlangen-Nürnberg, Erlangen, Germany
[3]Currently at Université Grenoble Alpes, CNRS, INRAE, IRD, Grenoble INP, IGE, Grenoble, France
[4]Laboratory of Hydraulics, Hydrology and Glaciology (VAW), ETH Zurich, Zurich, Switzerland
[5]Swiss Federal Institute for Forest, Snow and Landscape Research (WSL), Birmensdorf, Switzerland

**Correspondence:** Kaian Shahateet (k.fernandes@upm.es), Johannes J. Fürst (johannes.fuerst@fau.de)

**Abstract.** An accurate knowledge of the ice-thickness distribution on the Antarctic Peninsula Ice Sheet (APIS) is important to assess both its present and future responses to climate change. The aim of the present work is to improve the ice-thickness distribution map of the APIS by using a two-step approach. Such approach, which readily assimilates ice-thickness observations, considers two different rheological assumptions, and applies mass conservation in fast-flowing areas, where it also assimilates
ice-velocity observations. Using this method, we calculated a total volume of $27.7 \pm 10.1 \, 10^3$ km$^3$ for the APIS north of $70°$S. Using our ice-thickness map and the flux-gate method, we estimated a total ice discharge of $97.7 \pm 15.4$ km$^3$ a$^{-1}$ over the period 2015-2017, which is an intermediate value within the range of estimates by other authors. Our thickness results show relatively low deviations from other reconstructions on the glaciers used for validation. Qualitative analysis further reveals that our method properly reproduces the observed morphology of regional features, such as plateau areas, ice falls, and valley
glaciers. Despite the advances made in data assimilation and inversion modeling, further refinement of input data, particularly ice-thickness measurements, remains crucial to improve the accuracy of the APIS ice-thickness mapping efforts.

## 1   Introduction

The Antarctic Peninsula (AP) has undergone rapid changes in recent decades. The disintegration of Larsen A (1995) and Larsen B (2002) ice shelves on the east coast was attributed to hydrofracturing due to atmospheric warming (Rott et al., 1998; Scambos
et al., 2009; Banwell et al., 2013) and led to the acceleration of its tributary glaciers (Rott et al., 2011, 2018). Similarly, the disintegration of the Wordie Ice Shelf on the west coast accelerated the Fleming Glacier (Friedl et al., 2018), which has become dominant in terms of ice discharge from the AP (Shahateet et al., 2023).

Although the potential contribution of the Antarctic Peninsula ice sheet (APIS) to the sea-level rise ($69 \pm 5$ mm according to Huss and Farinotti, 2014) is small compared with that of the whole Antarctic Ice Sheet ($57.9 \pm 0.9$ m according to Morlighem
et al., 2020), it remains significant on decadal time scales due to the relatively short response time of its glaciers to environmental changes (Barrand et al., 2013). Additionally, the potential contribution of the AP to sea level rise is comparable to that



of the Canadian Arctic and significantly higher than that of regions such as High Mountain Asia, the Russian Arctic, Patagonia, or Alaska (Farinotti et al., 2019).

The mass loss in the APIS has increased from a mass change of $-6 \pm 13$ Gt a$^{-1}$ during the period 1997-2002 to $-35 \pm$
17 Gt a$^{-1}$ during 2007-2012, remaining thereafter at a similar level of $-33 \pm 16$ Gt a$^{-1}$ during 2012-2017 (Shepherd et al., 2018). Recently, Seehaus et al. (2023) calculated a geodetic mass balance (GMB) for the AP region north of 70°S, over the period 2014-2017, of $-24.1 \pm 2.8$ Gt a$^{-1}$. However, their study does not include most of the Palmer Land (drainage basins Ipp-J and Hp-I accordingly Rignot et al., 2013) area, which accounts for 60% of the glacierized area of the AP (Carrivick et al., 2018), making a direct comparison with the results of Shepherd et al. (2018) impractical. Nevertheless, it suggests an important
contribution to the mass loss from the whole APIS by its sector north of 70°S.

The estimation of ice discharge from the AP involves large uncertainties. Shahateet et al. (2023) calculated the APIS ice discharge north of 70°S using the five most commonly used ice-thickness map products for the AP. Their discharge revealed large differences for both individual outlet glaciers and for the entire region. Among the five models, three were interpreted to systematically underestimate the ice discharge from the AP due to a large number of zero and meaningless negative ice-thickness
estimations at the flux gates (Shahateet et al., 2023). The remaining two ice-thickness maps are not the most suitable choice for ocean-terminating glaciers, because they are physically-based methods employing either the shallow ice approximation (SIA) or the perfect plasticity (PP) approach. For these, thickness values are inversely proportional to local slopes, which tend to zero near the ice shelves and the terminus of tidewater glaciers. In the present work, our aim is to overcome this limitation.

Apart from ice-discharge estimates, accurately inferring the ice thickness is also important for predicting future glacier
evolution in response to climate change. In this case, reliable thickness estimates are required all over the glacierized areas - not only near the grounding line (Schannwell et al., 2015, 2016, 2018). For the Antarctic Peninsula North of 70°S (96,428 km$^2$), Huss and Farinotti (2014) calculated a total ice volume of 35,100 $\pm$ 3,400 km$^3$, giving a mean ice thickness of 364 $\pm$ 35 m. Although the study by Carrivick et al. (2018) covered a larger domain than that of Huss and Farinotti (2014), Carrivick et al. (2018) compared their results over the area where both studies overlap, resulting in a 26% larger ice volume of 44,164
km$^3$. In overlapping areas, differences are substantial. Also, in the same domain as Huss and Farinotti (2014) the models of Fretwell et al. (2013), Leong and Horgan (2020), and Morlighem et al. (2020), result in drastically different volume estimates of 22,815, 38,961, and 19,656 km$^3$, respectively. This calls for new studies that could reconcile the various estimates.

In this study, we aim to improve the knowledge of the ice-thickness distribution on the AP by adopting an approach similar to that of Fürst et al. (2017) to calculate the ice thickness for the APIS north of 70°S (Figure 1). We decided to center on this
area because it has been the focus of other important studies in this region, such as Huss and Farinotti (2014) and Seehaus et al. (2023), and because it is the most challenging in terms of ice-thickness reconstructions, due to its complex geometry. With this aim, we employed a two-step approach, whereby we initially used either the shallow ice approximation or the perfect plasticity as a first assumption. These two approaches are used in Huss and Farinotti (2014) and Carrivick et al. (2018), respectively, which are good approximations for plateau areas or wide valley glaciers with a certain slope, but may overestimate ice thickness
near the marine-terminating fronts, where basal sliding is larger and the surface slope is lower, since the equations of SIA and PP diverge to infinity when the slope approaches zero. The second step is addressed to solve these problems by updating the





ice thickness in fast-flowing regions using mass conservation in combination with regional velocity information. We obtained our final ice-thickness distribution by averaging the results from the second step of both the SIA and PP approaches.

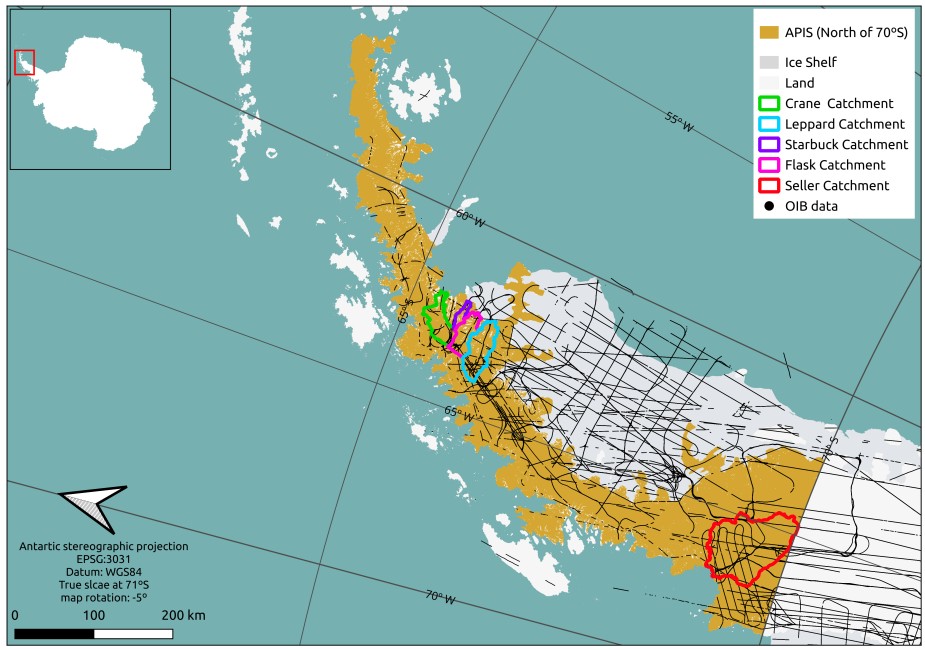

**Figure 1.** Study site with five glacier catchments of interest and radar flight lines of Operation IceBridge (OIB).

## 2 Data

The data required by the two-step ice-thickness reconstruction approach described in the Methods Section include glacier outlines, a digital elevation model (DEM), surface elevation change, surface mass balance (SMB), ice thickness, and surface velocity fields. In the following, we describe these data in more detail.

### 2.1 Glacier outlines

The glacier outlines used in this study are adapted from Silva et al. (2020). They applied a supervised classification method to
characterize glaciers in the AP region, specifically between latitudes 61°S and 73°S, including its periphery. The classification was primarily based on ASTER imagery, ASTER GDEM, and RAMP DEM. To ensure continuity in the first step of the reconstruction process, the outlines were merged to treat the glaciers of the Antarctic Peninsula as a single entity. We also used data from the SCAR Antarctic Digital Database, such as rock outcrops (Gerrish et al., 2020) to better define the glacier outlines.



Several glaciers in the AP have a plateau-like upper part that flows towards the ocean through cliffs that Gerrish et al. (2020) define as rock outcrops due to the presence of exposed rock. Our reconstruction (see methods) would interpret these as zero ice thickness, with no mass passing through these cliffs. Therefore, we manually removed such rock outcrops allowing glacier to flow across.

## 2.2    DEM

The digital elevation model (DEM) used in this study is the mosaic of the Reference Elevation Model of Antarctica (REMA) version 2 (Howat et al., 2022). REMA provides DEMs of the entire Antarctic continent at different resolutions. The elevation data in REMA is derived from satellite imagery, including WorldView-1, WorldView-2, WorldView-3, and GeoEye-1, using the SETSM software package (Howat et al., 2019), which employs stereography. For this study, the REMA mosaic with a 100-meter resolution was chosen to match the resolutions of other relevant thickness maps, e.g. Carrivick et al. (2018) and

Huss and Farinotti (2014).

## 2.3    Surface elevation change

The surface elevation change data used in this study were provided by Seehaus et al. (2023). They computed the average elevation changes in the Antarctic Peninsula region north of 70°S over the period 2013-2017. The elevation change data were derived using differential interferometric synthetic aperture radar (SAR) from the TanDEM-X satellite mission. The SAR data

were acquired during the austral winter to minimize the impact of differences in the penetration of the SAR signal into the snow/ice surface (Rott et al., 2018).

## 2.4    Surface mass balance

We obtained the SMB information for the 2008-2022 period from the regional climate model MARv3.12 ("Modele Atmosphérique Régional") with a spatial resolution of 7.5 km, expressed in mm of water equivalent per year (mm w.e. $a^{-1}$). It is a

regional 3D atmosphere-snowpack model coupled with the SISVAT (Soil Ice Snow Vegetation Atmosphere Transfer) model intended to simulate surface processes. Further information on the evaluation and parameterization of the MAR model on the AP can be found in Dethinne et al. (2023).

To use as input to our model, we calculated the average value of the SMB during the 2011-2020 period, with a resolution of 7.5 km, and downscaled it to a 100-m resolution using the REMA DEM, following a procedure similar to that of Huss and

Farinotti (2014), who applied a linear relationship between SMB and elevation. We used a sample size of 11x11 pixels of the original SMB data to compute a linear regression with respect to a resampled REMA DEM with the same resolution as the SMB (7.5 km). By applying now the linear regression to the original DEM (100 m resolution), we obtained a downscaled SMB with 100 m resolution. To obtain a smooth solution, we finally applied a moving average filter with a window size of 11x11 pixels on the downscaled SMB.





## 2.5 Ice-thickness measurements

The ice-thickness measurements used in this study were derived from data collected by Operation IceBridge (OIB; MacGregor et al., 2021) from NASA and the Alfred Wegener Institute (AWI; Braun et al., 2018a, b, c, d). OIB collected data from multiple instruments during the period 2009-2021. The AWI data specifically covers November 2014. The ice-thickness measurements were obtained using airborne ground-penetrating radar. Due to the large number of points provided by OIB and AWI, data reduction became necessary. Upon testing, we decided to keep only thickness measurements with a minimum distance of 500 m. Additionally, measurements with clear inconsistencies, such as thick ice in ice-free areas or on mountain ridges, were manually removed from the data set.

## 2.6 Velocity field

We used velocity data processed and provided by ENVEO (Environmental Earth Observation IT GmbH), as part of the Antarctic Ice Sheet Climate Change Initiative project of the European Space Agency (https://cryoportal.enveo.at/data/). It consists of monthly Antarctic ice velocity mosaics derived from Sentinel-1 Synthetic Aperture Radar (SAR) using feature-tracking techniques for the 2014-2021 period. The data are provided with 200-m resolution, in stereographic projection (EPSG:3031). The horizontal velocity components (in the x- and y-grid directions) are provided in true meters per day. From these data, we calculated a mean velocity field for the 2015-2017 period in the Antarctic Peninsula region, north of 70°S.

## 3 Methods

We based our ice thickness reconstruction on Fürst et al. (2017), which consists of a two-step inversion method implemented in Elmer/Ice, an open-source finite element software for modeling ice sheets, glaciers, and ice flow (Elmer/Ice, 2022). This approach applies shallow ice approximation and perfect plasticity in the first step, which gives a first estimate of the ice thickness. The aim of the first step using SIA is to calculate the ice flux ($F$) from the SMB ($\dot{b}_s$) and the local ice thickness change rate ($\partial H/\partial t$). In parallel to the SIA we used the PP approach based on Linsbauer et al. (2012), which assumes that the local stress in the glacier is equal to the material-specific yield stress, which requires only the glacier geometry as input (Perfect Plasticity). Since both assumptions are sensitive to slope, not being suitable for flat regions such as those near the glaciers termini, we then applied the second step, which updates the model in those areas with reliable velocity information available (see the flowchart in Figure 2). These regions overlap with the areas that are more prone to failure of the SIA and PP, therefore improving our reconstruction. Finally, both approaches were combined into a final thickness map by averaging the two results.





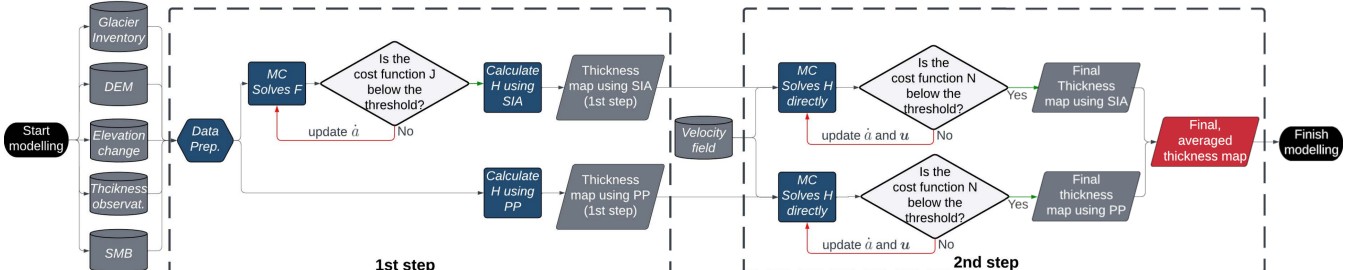

**Figure 2.** Flowchart depicting the ice-thickness reconstruction process. The first step involves calculating the ice thickness ($H$) using either the shallow ice approximation (SIA) or the perfect plasticity (PP) approach. In the SIA-based first step, the flux field ($F$) is determined by solving the mass conservation equation (Equation 2). The calculated flux field is then evaluated by the cost function ($J$) (Equation 3), which iteratively updates the apparent mass balance ($\dot{a}$) until the stopping criterion is met. At this stage, the flux map is utilized to calculate the ice thickness map ($H$) using the Shallow Ice Approximation (SIA) (Equation 4). Alternatively, the PP approach uses the glacier outlines and elevation change field to infer the ice thickness (Equation 5). In areas with reliable velocity observations ($|u| \geq 200$ m a$^{-1}$), we perform the second step. Each previously generated thickness map from the first step is used as a boundary condition to directly calculate $H$ by solving the mass conservation equation. The resulting ice-thickness map is then improved using the cost function ($N$) (Equation 6), which updates the values of $\dot{a}$ and $\boldsymbol{u}$. The process continues until the stopping criterion is met (i.e., the cost function falls below a given threshold), resulting in the ice-thickness maps using SIA and PP. Finally, the results of both approaches are combined by calculating the average value for each pixel, generating the final ice-thickness distribution.

## 3.1 First step

### 3.1.1 Shallow Ice Approximation

The method used in the present work to reconstruct the ice thickness (Fürst et al., 2017) is based on mass conservation.

Assuming that ice is incompressible, mass conservation can be formulated as (Cuffey and Paterson, 2010):

$$\frac{\partial H}{\partial t} + \nabla \cdot (\boldsymbol{u}H) = \dot{b}_{\mathrm{s}} + \dot{b}_{\mathrm{b}} \tag{1}$$

where $\frac{\partial H}{\partial t}$ is the local temporal change in ice thickness, $\boldsymbol{u}$ is the horizontal depth-averaged velocity vector, $H$ is the ice thickness, and $\dot{b}_{\mathrm{s}}$ and $\dot{b}_{\mathrm{b}}$ are the surface and basal mass balances, respectively. $\nabla \cdot$ is the 2D horizontal divergence. Rearranging Equation 1, we can obtain:

$$\boldsymbol{\nabla} \cdot \boldsymbol{F} = \dot{a} \tag{2}$$

where $\nabla \cdot \boldsymbol{F} = \nabla \cdot (\boldsymbol{u}H)$ is the two-dimensional flux divergence and $\dot{a}$ is the apparent mass balance ($\dot{a} = \dot{b}_{\mathrm{s}} - \partial H/\partial t$), which combines all sources and sinks, but neglects the basal mass balance ($\dot{b}_{\mathrm{b}}$).





The ice flux solution in the first step often shows high spatial variability and negative values. Therefore, to reduce undesirable characteristics, we iteratively update the apparent mass balance ($\dot{a}$), as a control variable, using a cost function $J$:

$$J = \lambda_{\text{pos}} \cdot \int_{\Omega} F^2 \int_{-\infty}^{-F} \delta(s) ds d\Omega + \lambda_{\text{reg}} \cdot \int_{\Omega} \left[ \left( \frac{\partial F}{\partial x} \right)^2 + \left( \frac{\partial F}{\partial y} \right)^2 \right] d\Omega + \lambda_{\dot{a}} \cdot \int_{\Omega} (\dot{a} - \dot{a}_{\text{init}})^2 d\Omega \tag{3}$$

where $\delta(s)$ is the Dirac delta function, with $s \in \mathbb{R}$. $\lambda_{\text{pos}}$, $\lambda_{\text{reg}}$, and $\lambda_{\dot{a}}$ are multiplier terms whose values were optimized in Fürst et al. (2017) to $10^2$, $10^0$, and $10^{-2}$, respectively, and $\Omega$ is the APIS domain. For the minimization of the cost function $J$, we rely on the module m1qn3 in Elmer/Ice (Gilbert and Lemaréchal, 1989), with a stopping criterion of $10^{-14}$, calculated as the ratio $|J_n - J_{n-1}|/J_{n-1}$, where $J_n$ and $J_{n-1}$ are the values of the cost function for the current and previous iteration. The first term on the right-hand side of Equation 3 penalizes solutions with negative flux, since the integral over $s$ of $\delta(s)$ with integration limits of $-\infty$ and $-F$ is zero if $F$ is positive and one if $F$ is negative. The integration over the APIS domain of $F^2$ multiplied by the integral mentioned above increases penalization for regions with more negative flux values. The second term increases with higher variability of $F$; in this way it favors smoother solutions. Finally, the third term penalizes the difference between the iteratively updated $\dot{a}$ and the initial apparent flux divergence input ($\dot{a}_{\text{init}}$), limiting the deviation of $\dot{a}$ from the inputs of SMB and $\partial H/\partial t$. Ultimately, the cost function $J$ is only a function of $\dot{a}$, since $F$ is also a function of $\dot{a}$ (Equation 2).

After the stopping criterion is reached, the ice flux is translated into ice thickness ($H$) in a post-process using the shallow ice approximation. The SIA assumes that the geometry of the glacier has a large aspect ratio (horizontal compared with vertical scales) (Hutter, 1983):

$$F^* = \frac{2}{n+2} B^{-n} (\rho g)^n \|\nabla h\|^n \cdot H^{n+2} \tag{4}$$

where $F^*$ is the corrected flux (see Appendix A), $B$ is the viscosity parameter, which is a priori unknown but can be calculated at points where ice-thickness measurements are available (e.g. flight measurements of Figure 1) using the same Equation 4 and the flux field calculated from the mass conservation described above. The viscosity parameter ($B$) is then spatially interpolated using bilinear interpolation over the entire domain. To minimize extrapolation artifacts, we implemented a new approach, in which we prescribed along the domain boundary a value of $B$ equal to the mean value of $B$ calculated over the points with ice-thickness measurements, previously removing the upper and lower 10% quantile. $\rho$ and $g$ are the ice density (917 kg m$^{-3}$) and the acceleration of gravity (9.18 m s$^{-2}$), respectively. Equation 4 assumes that there is only ice displacement due to deformation, neglecting basal sliding, limiting its application to slow-moving areas. We remove this constraint in the second step of the reconstruction, where we solve Equation 2 using the velocity information directly (without the need of the SIA). To avoid having Equation 4 diverging to infinity when solving for $H$, we set $\|\nabla h\| = \tan(1°)$ wherever the slope is less than $1°$.

### 3.1.2 Perfect Plasticity

Another approach, which uses only glacier geometry to infer the ice thickness ($H$) is the perfect plasticity approach:



$$H = \frac{\tau_d}{\rho g \tan \alpha} \tag{5}$$

where again $\rho$ and $g$ are the ice density and acceleration of gravity, respectively, $\alpha$ is the surface slope, and $\tau_d$ is the driving stress, which is a function of the material-specific yield stress ($\tau_0$). As in the case of SIA, we calculate $\tau_d$ at points with ice-

thickness measurements using Equation 5, and we then interpolate $\tau_d$ over the entire glacier domain, prescribing the mean value of $\tau_d$ of the entire domain (excluding the upper and lower quantiles 10%) at the domain boundaries, as done in the SIA approach. With the map of $\tau_d$ and surface slope (from the DEM), we can infer the ice thickness. Similarly to what we have done in the SIA case, in areas with $\alpha < 1°$, we set $\alpha = 1°$ to avoid having Equation 5 diverging to infinity.

### 3.2  Second step

In areas with reliable velocities, we perform the second step, in which the equation of mass conservation (Equation 1) is directly solved using all the inputs used in the first step, with the only additional information being the surface velocity. In this way, the only unknown variable in Equation 1 is the ice thickness $H$. Due to the fact that the velocity field ($\boldsymbol{u}$) in Equation 1 is a 2D vertically-averaged vector and the second step is applied only to fast flowing regions ($|u| \geq 200$ m a$^{-1}$), we opted to set the vertically averaged velocity to be equal to surface velocity. This assumption neglects internal deformation, which is not

unrealistic since in these regions basal sliding is expected to dominate (Bueler and Brown, 2009). We also used this velocity threshold because remote sensing-derived velocity fields usually contain gaps, and the uncertainty can dominate the signal in slow-moving areas.

In the second step, the domain of fast flowing regions (a sub-domain of the glacier outline) requires lateral boundary conditions, which come from the first step either from SIA or the PP solution fields. We set the ice thickness calculated from the

first step as a Dirichlet boundary condition along the lateral margins of the fast-flowing domain. No boundary conditions are imposed on the marine ice front.

Once we get the ice-thickness map from the second step by solving the mass conservation equation, we use a cost function similar to Equation 3, since we cannot ensure that the input data are consistent in terms of mass balance:

$$
\begin{aligned}
N =\, & \gamma_{\text{pos}} \cdot \int_{\Omega} H^2 \int_{-\infty}^{-H} \delta(s) ds d\Omega + \gamma_{\text{reg}} \cdot \int_{\Omega} \left[ \left(\frac{\partial H}{\partial x}\right)^2 + \left(\frac{\partial H}{\partial y}\right)^2 \right] d\Omega \\
& + \gamma_{\dot{a}} \cdot \int_{\Omega} \left( \dot{a} - \dot{a}^{\text{init}} \right)^2 d\Omega + \gamma_{\text{U}} \cdot \sum_{i=1}^{2} \int_{\Omega} \left( u_i - u_i^{\text{init}} \right)^2 d\Omega
\end{aligned}
\tag{6}
$$

As in the previous cost function (Equation 3), the first and second terms in Equation 6 are penalizations for negative and highly variable ice thickness values, respectively. The third and fourth terms are intended to maintain apparent flux divergence ($\dot{a}$) and velocity ($u_i$) close to their initial input values. Again, $\gamma_{\text{pos}}$, $\gamma_{\text{reg}}$, $\gamma_{\dot{a}}$, and $\gamma_{\text{U}}$ are the multipliers of their respective terms. We chose to use the same values as Fürst et al. (2017) of $10^2$, $10^{-2}$, $10^{-4}$, and $10^{-8}$, respectively. The cost function





$N$ iteratively updates the values of apparent flux divergence and the velocity field until the same stopping criterion scheme of
Equation 3 is met. If Equation 6 is solved without further modifications, the iterative optimization preferentially updates $\dot{a}$ due
to the different magnitudes of the velocity field and the apparent flux divergence. To better align the relative changes, we scaled
down the velocity derivative by a factor of 0.05.

Once the second step is completed, the results are two ice-thickness maps: one based on SIA and another on PP, but both of
them have the fast-flowing regions updated using velocity information. To have a single ice-thickness map, we then calculate a
final ice-thickness map by computing the mean value of the individual pixels of the SIA and PP reconstructions. This averaging
of two different models is supported by the intercomparison of Farinotti et al. (2017, 2021), where they showed the advantages
of ensemble models. Finally, the uncertainty estimation of the various steps is presented in Appendix B.

## 4   Results

Our results in Table 1 indicate that the total volume of ice in the APIS north of $70°$ is $27.4 \pm 9.1$ or $27.9 \pm 9.1$ $10^3$ km$^3$,
according to the assumption, SIA or PP respectively. Considering an area of $81,535$ km$^2$, we get mean ice thicknesses of
336 and 342 m, respectively. The average ice-thickness distribution of SIA and PP (using the two steps) is shown in Figure
3. Furthermore, using the same methods and flux gates of Shahateet et al. (2023) (see Appendix C) resulted in ice-discharge
values of $95.4 \pm 11.8$ and $99.7 \pm 11.8$ km$^3$ a$^{-1}$, respectively.

Table 1 also shows the impact of the second step on the ice discharge calculation. The ice discharge is more sensitive than the
ice volume to the use of the second step, since the latter is only applied to areas with large velocities, which include the zones
upstream of the grounding lines and marine termini, where the ice discharge is calculated. Regarding the changes implied by
the introduction of the second step, the total volume decreased by 1.1% and by 0.4% when SIA and PP were used, respectively.
In the calculation of ice discharge, the introduction of the second step implied a decrease by 16.4% in the case of SIA and an
increase by 1.5% in the case of PP.

As the two assumptions (SIA and PP) led to similar results, we combined them in a final solution by calculating their mean
value for each pixel. This resulted in a total volume of $27.7 \pm 10.1 \times 10^3$ km$^3$ and an ice discharge of $97.7 \pm 15.4$ km$^3$ a$^{-1}$.
The discussion presented hereafter will refer to the results combining the SIA and PP approaches.

**Table 1.** Total volume and ice discharge calculated using the two approaches (SIA and PP) and the combination of both, using the first step
only, and using the two steps. See appendices B and C for details about error estimation.

| steps applied | Total volume ($10^3$ km$^3$) | | | Ice discharge (km$^3$ a$^{-1}$) | | |
|---|---|---|---|---|---|---|
| | SIA | PP | combined | SIA | PP | combined |
| 1$^{st}$ only | $27.7 \pm 9.5$ | $28.0 \pm 9.5$ | $27.8 \pm 10.6$ | $114.1 \pm 29.3$ | $98.2 \pm 29.4$ | $105.8 \pm 41.1$ |
| 1$^{st}$ and 2$^{nd}$ | $27.4 \pm 9.1$ | $27.9 \pm 9.1$ | $27.7 \pm 10.1$ | $95.4 \pm 11.8$ | $99.7 \pm 11.8$ | $97.7 \pm 15.4$ |

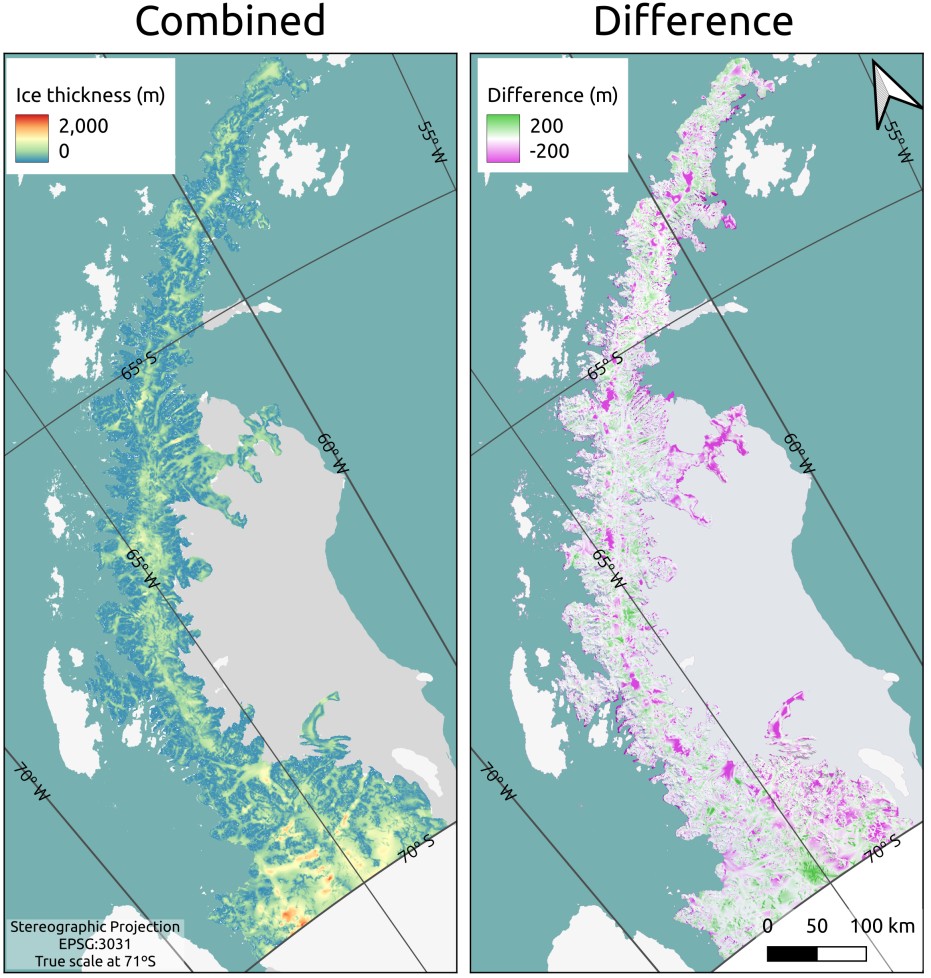

**Figure 3.** Ice-thickness reconstructions using a 2-step approach: Combined - average thickness of both approaches; Difference - difference of the result using Shallow Ice Approximation (SIA) and Perfect Plasticity (PP) using the two steps. Color of the difference scale has 60% transparency to enable the visualization of the optical image behind.



## 5 Discussion

### 5.1 Discussion on the methodology

In general, the first step of the SIA and PP approaches produces ice-thickness reconstructions consistent with the relief of our study site (ridges, valley glaciers, and ice caps on the AP plateau). We adjusted the glacier outlines by manually connecting the plateau ice to its outlet glaciers (cf. Section 2), as otherwise the ice thickness would be underestimated. The reason is that such unrealistic internal boundaries would imply zero-flux conditions and hence zero ice thickness.

The first step using the SIA neglects basal sliding, which leads to an overestimation of the ice thickness in fast-flowing

regions (see the differences in Figure 3). In contrast, the introduction of the second step decreases the ice-thickness values in fast-flowing regions when the SIA approach is used. Furthermore, SIA is sensitive to slope, specifically in flat areas near the terminus and on the plateau (see Equation 4). The second step then improves the reconstruction in those regions, in particular near the marine termini. When the second step is applied to the PP approach, the ice discharge increases slightly. Although the PP approach is also sensitive to slope, it does not make any assumption about the velocity distribution throughout the glacier.

### 230 5.2 Comparison against previous regional-scale estimates

We now compare our results with the ice-thickness maps most often used in the literature, namely Carrivick et al. (2018), Huss and Farinotti (2014), Fretwell et al. (2013), Leong and Horgan (2020), and Morlighem (2022). An important application of ice-thickness reconstructions is the computation of ice discharge to the ocean. By combining the latter with the climatic mass balance, an estimate of the total mass balance is obtained (e.g. Rignot et al., 2019; Shepherd et al., 2018). Therefore,

we focus on comparing the ice discharge estimates obtained when using the different ice-thickness data sets. It is important to remark that, as done in Shahateet et al. (2023), the ice discharge computations are in all cases performed using the same set of flux gates and a common velocity field, so the differences in ice discharge between methods can be solely attributed to the differences in ice thickness at the flux gates. To analyze the differences in results between pairs of models, we used the equation introduced by Shahateet et al. (2023), where $\Delta\varphi_{jk}$ represents a normalized mean difference in the calculation of ice

discharge using models $j$ and $k$ for the whole set of flux gates:

$$\Delta\varphi_{jk} = \overline{\left[ \frac{|\varphi_{i,j} - \varphi_{i,k}|}{\frac{1}{6}\left(\sum_{l=1}^{6} \varphi_{i,l}\right)} \right]} \tag{7}$$

where the fraction on the right-hand side represents the difference in the absolute value of the ice discharge calculated using the models $j$ and $k$ for the flux gate $i$, normalized by the mean of all models for that gate (the term in the denominator). The global average bar is understood to be applied over the whole set of flux gates (represented by the subscript $i$). The differences

between our estimate of ice discharge and those of other authors are summarized in Table 2.

Table 2 shows that the results can be grouped into two different families. On the one hand we have the physically-based approaches (namely Huss and Farinotti, 2014 and Carrivick et al., 2018), with which our model has lower differences in





**Table 2.** Mean of the normalized difference of ice discharge between our model and other models (see Equation 7).

| Carrivick | H&F | Bedmap2 | DeepBedMap | Bedmachine v3 |
|-----------|-----|---------|------------|----------------|
| 0.63 | 0.53 | 1.12 | 1.21 | 1.12 |

terms of ice discharge of the individual basins, and on the other hand we have the neural-network and interpolation-based approaches (namely Leong and Horgan, 2020; Fretwell et al., 2013; Morlighem, 2022) with larger differences in comparison

with our results. Note that although Morlighem (2022) use mass conservation to infer the ice thickness of the entire Antarctica, in the APIS they mainly used anisotropic diffusion, so we classified their APIS reconstruction into the interpolation-based group. When considering the total ice discharge estimate (Table 3), our model result is in between the two families, with the difference in total discharge between our model and H&F being similar to the difference between our model and Bedmap2 (and DeepBedMap), and the difference between our model and Carrivick's being similar to the difference between our model

and Bedmachine's.

**Table 3.** Total ice volume and ice discharge calculated in the present work making use of the two-step approach considering shallow ice approximation (SIA) and perfect plasticity (PP), combined into a final, single ice-thickness field, and the results obtained using the different ice-thickness models available in the literature for the Antarctic Peninsula north of $70°$ S (Carrivick et al., 2018 (Carrivick), Huss and Farinotti, 2014(H&F), Fretwell et al., 2013 (Bedmap 2), Leong and Horgan, 2020 (D.BedMap), and Morlighem, 2022 (B.machine v.3)). Carrivick et al. (2018) and Leong and Horgan (2020) do not provide error estimates of their ice-thickness reconstructions.

| Parameter | Present work (two steps) | Others | | | | |
|-----------|--------------------------|--------|-----|--------|----------|-----------|
| | Combination of SIA and PP | Carrivick | H&F | Bedmap | D.BedMap | B.machine |
| Volume ($10^3$ km$^3$) | $27.7 \pm 10.1$ | 38.7 | $29.3 \pm 7.6$ | $19.0 \pm 13.6$ | 32.4 | $18.1 \pm 3.2$ |
| Discharge (km$^3$ a$^{-1}$) | $97.7 \pm 15.4$ | 140.7 | $128.2 \pm 31.6$ | $62.7 \pm 59.4$ | 59.1 | $54.9 \pm 14.9$ |

We also observe in Table 3 that our model produces a total ice volume very close to that of Huss and Farinotti (2014), which is also the one showing the closest ice discharge to our own estimate. Comparing the ice discharge of our first step reconstruction using SIA (Table 1), which is an approach similar to that used by Huss and Farinotti (2014), based on mass conservation, results in an 11% lower discharge. Combining the approaches of SIA and PP, both updated by the second step,

produced an ice-discharge value 24% lower than that of Huss and Farinotti (2014), while our total ice volume was 5.5% lower than that of Huss and Farinotti (2014), and did not change significantly when the second step was introduced.

The differences are larger with respect to the reconstruction of Carrivick et al. (2018). Our first step using the PP approach resulted in an ice discharge and total volume 30% and 28% lower, respectively, than those of Carrivick et al. (2018). When the second step was performed and the two reconstructions were combined, the differences to Carrivick et al. (2018) remained

almost equal to those of the first step using PP, with ice discharge and total ice volume 32% and 26% lower, respectively. Although the results by Carrivick et al. (2018) are based on PP, their calculation of the driving stress ($\tau_d$) is different. Whereas



we calculate $\tau_d$ by inversion of Equation 5 and then interpolation throughout the domain, Carrivick et al. (2018) used an empirical relation proposed by Driedger and Kennard (1986) based on curve fitting between area, slope, and basal shear stress using data of the major glaciers of four Cascade Range volcanoes in North America, with areas ranging from 1.2 to 11.0 km$^2$,

which have markedly different geometries compared to the AP glaciers. They then validated their fit using glaciers from other regions (which did not include glaciers in the AP) ranging from 1.0 to 4.1 km$^2$. These characteristics of the glacier used by them contrast with glaciers in the APIS north of 70°. In this region, the glaciers range from 2.9 to 7000 km$^2$, with a mean area of 100 km$^2$. Therefore, the empirical relation proposed by Driedger and Kennard (1986) extrapolates most of the basal shear stress in our region of interest (63% of the glaciers in the region are greater than 11 km$^2$). Another characteristic of the

empirical relationship that can lead to an overestimation of the basal shear stress in the APIS is the fact that the elevation range in volcano glaciers are usually greater than the elevation range of the APIS glaciers, resulting in relatively fewer elevation bands (see Carrivick et al., 2018), implying even larger areas of the individual elevation bands. Furthermore, implicit in their assumption is the fact that larger glacier areas correspond to thicker ice (which would result in greater basal shear stress): But in the case of the APIS glaciers, this could be achieved through a different relationship to that of Driedger and Kennard (1986),

given the completely different geometry of both sets of glaciers.

Regarding the comparison with the neural network and interpolation-based models of tables 2 and 3 (Leong and Horgan, 2020; Fretwell et al., 2013; Morlighem, 2022), our results are substantially larger for both ice volume and ice discharge, except for the ice volume calculated by Leong and Horgan (2020), which is similar to our own estimate. The largest difference in the calculation of the ice discharge ($\Delta\varphi$) with respect to our model is that of the Bedmachine model.

Shahateet et al. (2023) noted that Bedmachine v2 model (Morlighem et al., 2020) used mainly anisotropic diffusion to estimate the ice thickness in the Antarctic Peninsula, except for three glaciers, namely the Flask, Leppard, and Seller glaciers. This anisotropic diffusion is a type of interpolation in which the quality of the reconstruction can be highly impacted by the lack or scarcity of ice-thickness measurements (which is also the case for Bedmap2, since it is a pure interpolation method). Shahateet et al. (2023) highlighted that Bedmachine v2 shows 24% near-zero ice-thickness values (a situation close to that of Bedmap2)

along the flux gates that they used to calculate the ice discharge of the APIS. This explains their low ice discharge values. Indeed, when we compare our ice-discharge estimates with those of Bedmachine v3 restricting to the three glaciers where they applied mass conservation, the differences decrease substantially. The normalized difference in ice-discharge calculation between these two models (see Equation 7) for the Flask, Leppard, and Seller glaciers are 0.45, 0.29, and 0.22, respectively. These differences in ice-discharge calculation contrast markedly with the value of 1.12 obtained when all APIS glaciers are consid-

ered (Table 2). This shows that when Bedmachine v3 is applied, in fast flowing glaciers, using mass conservation produces ice discharge estimates closer to our own ones.

Focusing now on the differences with DeepBedMap, we note that they used a super-resolution deep convolutional neural network trained with Antarctic data external to the AP, which may affect the final solution in our region of interest. Furthermore, Shahateet et al. (2023) demonstrated the high variability and the large number of negative ice-thickness values in the

DeepBedMap reconstruction of the APIS (we masked such negative ice-thickness values to zero thickness values in our calcu-



lations using their model). Finally, Figures 6 and D1 show that the three mentioned reconstructions (Bedmap2, DeepBedMap and Bedmachine) may underestimate the ice thickness in the valley glaciers of the AP.

Regarding the spatial distribution of the differences in ice-discharge calculated using the different models, Figure D2 shows the multi-model normalized root-mean-square deviation (Shahateet et al., 2023) among the different models. As in Shahateet

et al. (2023), it illustrates the inhomogeneous spatial distribution of the differences. In particular, it is remarkable the higher similarity in results for the glaciers terminating in ice shelves as compared with those terminating in the ocean. Shahateet et al. (2023) attributed this unequal spatial distribution to the fact that there are more abundant radar flight lines on glaciers that end on ice shelves, thus better constraining the models. This highlights the importance of undertaking further ice-thickness measurements to reduce the uncertainty in the calculated ice discharge and total volume.

**5.3    Comparison with radio-echo sounding data**

To validate the performance of the different models at the single-glacier scale, we used radio-echo sounding (RES) data from Farinotti et al. (2013) on the Flask and Leppard glaciers acquired in 2010 and 2011. Farinotti et al. (2013) used an ice-flow model to reinterpret the RES data, categorizing the ice-thickness measurements into clear, easy-to-interpret signal and very diffuse, speculative signal. Farinotti et al. (2013) also identified signals that are probably reflections from the bed and signals

that are probably reflections from the side walls. With the aim of comparing the different models available for the APIS north of 70°S, we calculated the normalized mean absolute error (NMAE, see Shahateet et al., 2023) of the individual models with respect to the RES measurements of Farinotti et al. (2013), using only their better-quality P01, P02 and P03 ice-thickness measurements (see Figure 4a).

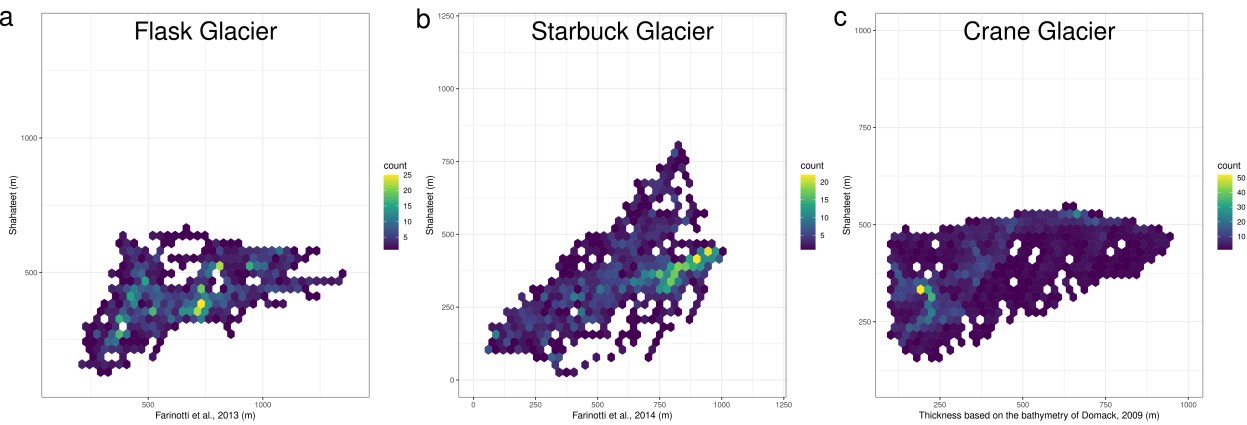

**Figure 4.** Hexbin plot of three different sources compared to our reconstruction: a) radio-echo sounding (RES) of Flask and Leppard glaciers from Farinotti et al. (2013), b) RES of Starbuck Glacier from Farinotti et al. (2014), and c) inference of ice thickness using DEM from REMA, bathymetry from Domack (2009), and flotation criterion.



The lowest NMAE of all six reconstructions corrresponds to Bedmap2, with 29%. The reconstruction of Bedmachine v3 and
ours had similar NMAEs of 33 and 35%, respectively. DeepBedMap, Carrivick, and H&F had similar values of 58%, 60% and
66%, respectively. Due to the easy access to the input data used by Bedmap2 (see Fretwell et al., 2013), we can analyze the
reason why Bedmap2 has a lower error than our reconstruction on the Flask Glacier. Both inversions used the OIB data from
2011, but Fretwell et al. (2013) managed to identify data from ice thickness measurements with high uncertainty, excluding
them from their reconstruction. Therefore, when they used their method to interpolate the ice thickness, they had available
good data overlapping the same region where Farinotti et al. (2013) had high-quality data.

Although the reanalysis by Farinotti et al. (2013) constitutes the most reliable validation data for the ice thickness in the
AP available in the literature, their analysis covers only the Flask and Leppard glaciers, forcing us to use other resources to
validate the models. These include RES measurements on the Starbuck Glacier from 2012 (Farinotti et al., 2014) (Figure 1).
These data were used as input to the ice-thickness inversion of Huss and Farinotti (2014), so they cannot be used to validate
their bed reconstruction. The velocity field on the Starbuck Glacier used in the present work is below 200 m a$^{-1}$; consequently,
for this specific glacier we only performed the first step of our two-step approach.

Figure 4b shows the hexbin chart of the thickness of Farinotti et al. (2014) and the present reconstruction. Compared to the
other reconstructions, Carrivick's and our solution had similar NMAEs with respect to the RES measurements with 37% and
42%, respectively. Bedmap2, DeepBedMap, and Bedmachine v3 had values of 73%, 63%, and 91%, respectively. The NMAE
of H&F with respect to the RES measurements, of 27%, cannot be considered for validation because, as mentioned earlier,
such data were used for calibration of their ice-thickness reconstruction. But this NMAE helps to highlight the relatively good
performance of Carrivick's and our own reconstruction.

The independent data on Flask and Starbuck glaciers used to validate the models highlights the marked errors in all recon-
structions, but shows that the present work is among the reconstructions presenting lower errors for the two glaciers analyzed.

Due to the lack of other reliable field measurements, we used indirect measurements of ice thickness to evaluate the recon-
structions. In the following, we use bathymetric data together with a flotation criterion to estimate the ice thickness of the Crane
Glacier. According to Needell and Holschuh (2023), the Crane glacier (see location in Figure 1) experienced a retreat of more
than 10 km after the collapse of the Larsen B ice shelf during 2002-2004. After that, the glacier front was relatively stable until
2010, when the glacier started to readvance until 2021. In 2006 (during the stable period with the glacier retreated), Domack
(2009) acquired bathymetric data from the Crane Glacier Embayment. With the readvance of Crane Glacier, this bathymetry
became bed topography near the glacier terminus. Using these data together with surface elevation from REMA, we infered
the thickness using a flotation criterion. In the raster cells where the surface elevation ($h$) was $< \frac{1}{9}$ of the seafloor depth ($d$), we
assumed that the ice was floating (thus $H = 10h$, where $H$ is the thickness and $h$ is the surface elevation). On the other hand,
if the surface elevation was $\geq \frac{1}{9}$ of the seafloor depth, then we set $H = h + d$.

Figure 4c shows the hexbit chart of this reconstruction of ice thickness in the front of the Crane Glacier and our 2-step
reconstruction. Again, we used the NMAE to evaluate the performance of the reconstructions. The lowest NMAE was found
for Bedmap2, with a NMAE of 46%, followed by our reconstruction with 66%. Carrivick, H&F, and Bedmachine v3 had
NMAEs of 72%, 79%, and 99%. The NMAE of Bedmachine v3 was close to 100% due to the large amount of zero ice-





thickness values. Finally, DeepBedMap resulted in an NMAE of 183% without removing the negative values of ice-thickness
and 94% when the negative values were set to zero (see Shahateet et al., 2023 for a complete discussion of the negative
ice thickness values of the DeepBedMap reconstruction). Part of the large NMAE can be attributed to the error in the DEM
data, since every meter of uncertainty of the DEM will be multiplied by 10 when we apply the flotation criterion. Another
important contributor to the uncertainty of the ice-thickness reconstructions of this specific glacier is the fact that ice-thickness
measurements of the OIB were acquired in 2002, 2004 and 2016. Therefore, reconstructions made before 2016 only used OIB
data from 2002 and 2004, which according to Needell and Holschuh (2023) was the period of retreat of Crane Glacier. This
contrasts with the data acquisition period to mosaic the DEM of REMA (2012-2020), when the glacier was advancing. The
REMA DEM was used to calculate the ice thickness of Crane Glacier assuming the flotation criterion, which would reflect the
ice thickness of its acquisition period. Even in the reconstruction proposed here, our algorithm (Section 2) selected the OIB
data from 2002, which precisely are the same data used by Fretwell et al. (2013), and these are the two models with lowest
NMAE in comparison with the ice thickness calculated using the flotation criterion. We therefore attribute the large error to
the high uncertainty of the input data of the models.

The reanalysis of Farinotti et al. (2013) mentioned previously also allows us to evaluate the quality of the ice-thickness
measurements used in our inversion, which in the Flask region consists of only OIB data. Figure 5 highlights the differences
in the Flask Glacier between the OIB measurements and the reanalysis of Farinotti et al. (2013). Some points just 10 m apart
in OIB and Farinotti et al. (2013) data differ in ice thickness by more than 100% (e.g. within 10.5 m, OIB measured a value
of 1562 m, while the reanalysis of Farinotti et al., 2013 provided a value of 738 m). Even the OIB measurements sometimes
lack self-consistency. In a specific case, for example, the OIB provided ice-thickness values of 1637 and 761 for measurements
points just 6 m apart. Other inconsistencies appear in the OIB data. For instance, there are measurements of thick ice (more
than 500 m) over rock outcrops. Another zone (on the side mountains of Hariot Glacier) with steep slope had measurements of
2200 m thick. All of these inconsistencies result in unrealistic ice viscosity and driving stress, in equations 4 and 5, respectively,
which especially affect our first step estimates.

### 5.4 Qualitative indicators of performance

Qualitatively, it is also possible to evaluate how well a given ice-thickness reconstruction reproduces the features seen in a
certain region. For example, the APIS region has some characteristic features, such as a plateau in the spine of the peninsula
that accumulates ice, which then flows towards the ocean often through very steep slopes and cliffs, frequently forming ice
falls, and then the slopes get gentler in the outlet valley glaciers. We selected two regions with markedly different amounts of
field data for qualitative analysis. The region in Figure 6 was selected due to the large amount of ice-thickness measurements
available (as seen in the previous analyses of Crane, Starbuck, Flask, and Leppard glaciers), which provide more constraints to
the reconstructions. By contrast, we selected the region in Figure D1 (northern tip of the AP) due to the lack of measurements,
which resulted in poor constraints to the reconstructions.

In Figure 6, despite the thickness differences, we can see that all three reconstructions (Carrivick, H&F and ours) reproduce
thick ice over the plateau, thinning toward the cliffs (due to increasing slope, thus accelerating the ice) and then thickening



**Figure 5.** Ice-thickness measurements on Flask Glacier (see Figure 1 for location). The first panel shows data from Operation IceBridge (OIB), while the second panel shows the reanalisys of Farinotti et al. (2013). Lastly, the third panel superimposes the two previous panels. The number of measurement points along lines was reduced to allow a better visualization.





**Figure 6.** Close-up of the reconstructions of Carrivick, H&F, the one of the present work, Bedmap2, DeepBedMap, and Bedmachine v3 on the region surrounding the Crane, Starbuck, Flask and Leppard glaciers, all of which flow towards the embayment of the former Larsen B ice shelf. Color of the ice-thickness scale has 60% transparency to enable the visualization of the optical image behind. We masked ice-thickness values lower than 10 m.



again in the valleys, with a flatter slope. The other three reconstructions (Bedmap2, DeepBedMap, and Bedmachine v3) shown in Figure 6 also reproduce thicker ice on the plateau and a thinning near the cliffs, but do not reproduce some glaciers in the

valley regions. In the case of DeepBedMap, a highly-pixelated reconstruction is also shown.

In Figure D1, we see again that Carrivick, H&F and our model reconstruct thick ice on the plateau and a thinning when it passes through the cliffs, thickening when the valleys are reached. Bedmap2 and DeepBedMap also reproduced thicker ice on the plateau thinning toward the cliffs, again failing to reproduce the glaciers in the valleys. However, Bedmachine v3 did not properly reproduce the ice thickness in the northernmost part of this region, where it estimates ice thicknesses lower than 10

m.

## 6    Conclusions and outlook

In the present work, we provide a new ice-thickness map of the Antarctic Peninsula Ice Sheet north of $70°$ S. This map was constructed based on the two-step approach proposed by Fürst et al. (2017). The first step estimates the ice-thickness distribution using two different approaches, namely the shallow ice and the perfect plasticity approximations. For fast-flowing areas, the

second step updates the ice-thickness from the first step using mass conservation, with the aim of overcoming the limitations of SIA and PP near the glacier termini, where such approaches do not perform properly due to the low slopes. Finally, the results from both approaches are combined into a single ice-thickness product. Using this product, we estimated a total volume of $27.7 \pm 10.1 \ 10^3$ km$^3$ for the APIS north of $70°$ S, and a total ice discharge of $97.7 \pm 15.4$ km$^3$ a$^{-1}$ over the period 2015-2017, a value that lies in the middle range of the estimates by other authors, which show a large spread.

Comparing our results with those summarized in Shahateet et al. (2023) for the same region showed that our ice-thickness reconstruction produces results similar to those of Huss and Farinotti (2014) in terms of spatial distribution of ice thickness, total volume, and ice discharge (especially when only SIA is applied) and comparable results to those of Carrivick et al. (2018) in terms of spatial distribution of ice thickness. On the other hand, our solution showed substantial differences from those of Fretwell et al. (2013), Leong and Horgan (2020) and Morlighem (2022). We attribute these similarities and differences to the

use of physically-based approaches by our model and those of Huss and Farinotti (2014) and Carrivick et al. (2018), while Fretwell et al. (2013) and Morlighem (2022) are interpolation-based approaches and Leong and Horgan (2020) is a super-resolution, neural network-based approach based on Fretwell et al. (2013), all of which are highly affected by the scarcity of ice-thickness measurements. Furthermore, we showed that the reconstructions of H&F, Carrivick, and ours qualitatively reproduced the features expected in the region, including thick ice on the plateau areas of the APIS, thinner ice on the high-

slope areas in the transition zones from the plateau areas to the outlet glaciers draining them, and then thicker ice on the outlet valley glaciers. Bedmap2, DeepBedMap, and Bedmachine v3, however, failed to reproduce some of these features.

All models showed large errors when validated against independent ice-thickness data. In the case of the most reliable set of independent data, namely those of Farinotti et al. (2013) on Flask Glacier, errors ranged from 29% to 66%. This highlights the need for more dense and accurate field measurements of ice thickness in the APIS region, as proposed by the RINGS action



group of SCAR. This is needed to reduce the uncertainties in total ice discharge and total volume, which have an impact on the projections of future contributions to sea-level rise from ice wastage from this region.

Regarding the currently available ice-thickness data, remarkable inconsistencies among them are sometimes present. Some of them could derive from improper positioning data of the flight lines and some from the presence of temperate ice, the latter especially in the northernmost regions. The usefulness of the available ice-thickness data could be improved by reanalyses such
as the one performed by Farinotti et al. (2013), involving a detailed quality check of the radio-echo sounding data leading to a classification of the measurements into a range from clear, easy-to-interpret signal to very diffuse, speculative signal. Also, reprocessing some problematic OIB flight lines could be beneficial to reduce uncertainties.

As the first step preserves the ice-thickness observations, measurement errors were directly translated into our ice-thickness maps. The second step was able to update the ice thickness in areas with velocities greater than 200 m a$^{-1}$. However, in slow-
moving areas, the ice thickness was not updated, preserving the measurement errors. In the future, this could be refined by applying a cost function to the viscosity and driving stress fields, taking into account their absolute value and their variability, as well as including the temperature dependency of the viscosity.

## 7    Data availability

The ice thickness map of the present work can be found at https://gitlab.com/kaian_shahateet/apis-ice-thickness-reconstruction.
git

## 8    Author contribution

Conceptualization, K.S., J.J.F., T.S., and F.N.; methodology, K.S. and J.J.F.; software, K.S. and J.J.F.; formal analysis, K.S., J.J.F., and F.N.; data curation, K.S., T.S., and D.F.; writing-original draft preparation, K.S. and F.N.; writing-review and editing, J.J.F., T.S., D.F., and M.B.; visualization, K.S.; supervision, F.N., T.S., and M.B.; funding acquisition, F.N. and M.B. All authors
have read and agreed to the published version of the manuscript.

## 9    Competing interests

Some authors are members of the editorial board of The Cryosphere Journal.

## 10    Acknowledgements

This research was funded by the innovation programme via the European Research Council (ERC) as a Starting Grant (FRAG-
ILE project) under grant agreement No. 948290 and by grant PID2020-113051RB-C31 from MCIN / AEI / 10.13039 / 501100011033 / FEDER, UE. KS was funded by the German Academic Exchange Service (DAAD) under grant 91828107. TS was funded by the ESA Living Planet Fellowship MIT-AP and the Elite Network of Bavaria grant IDP M3OCCA.



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

# Appendices

## A. Flux-based solution

The zero transitions in the flux field would be transmitted to the ice-thickness solution. As in Fürst et al. (2017), we also avoid that by correcting the flux field as:





$$F^* = (1 - \kappa) \cdot \|F\| + \kappa \cdot F_{\text{crit}} \,, \text{ where}$$


$$\kappa = 1 - 2/\pi \cdot \arctan\left(F^2/F_{\text{crit}}^2\right)$$

(8)

where, $F_{crit}$ is defined as 10% of the mean flux magnitude across the entire domain.

## B. Error estimation of the ice-thickness reconstruction

We first calculate the error in the first step using the shallow ice approximation (and assume the same error for the perfect plasticity case), and then update the error values in areas where we performed the second step.

**B.1 Estimation of error in the first step**

We calculate the error map in the first step (valid for both SIA and PP) using the SIA equations. The error in the ice-thickness distribution stems from the uncertainty in the input fields of SMB and $\partial H/\partial t$. As Fürst et al. (2017), we assume that the errors are transmitted through the mass conservation equation (Equation 1) and are then scaled by the SIA when deriving the ice thickness from the flux (Equation 4). We also assume that the inaccurate flux field ($F + \delta F$) also satisfies the mass

conservation equation (Morlighem et al., 2014):

$$\nabla \cdot [(F + \delta F)(\boldsymbol{n} + \delta \boldsymbol{n})] = \dot{a} + \delta \dot{a} \tag{9}$$

where $F$, $\boldsymbol{n}$, and $\dot{a}$ are the flux, the flux direction, and the apparent mass balance, respectively, and the variables preceded by $\delta$ denote their corresponding errors. Neglecting second-order terms and accounting for the fact that $\nabla \cdot \boldsymbol{F} = \dot{a}$, results in

$$\nabla \cdot [\boldsymbol{n}\delta F] = \delta\dot{a} - \nabla \cdot [F\delta\boldsymbol{n}] \tag{10}$$

Along the land-terminating margin, we assume zero flux, and thus the error in flux is zero at these locations. The error in the ice-thickness measurements also contributes to the error in our reconstruction. We assume that at the measurement locations, the flux is known with an uncertainty that is proportional to the thickness measurement uncertainty. Therefore, the reported error in the measurements is converted into its equivalent in error in flux using Equation 4 without the correction of Appendix A. Equation 10 does not take into account that the flux is constrained upstream of each measurement point. For this reason, we

assume that the error propagates once downstream ($\delta H_1$) and once upstream ($\delta H_2$), converting Equation 10 into two equations:

$$\nabla \cdot [(+\boldsymbol{n})\delta F_1] = \delta\dot{a} + \|\nabla \cdot [F\delta\boldsymbol{n}]\|$$

$$\nabla \cdot [(-\boldsymbol{n})\delta F_2] = \delta\dot{a} + \|\nabla \cdot [F\delta\boldsymbol{n}]\| \tag{11}$$



The above pair of equations is structurally identical to Equation 1 and is numerically solved as described in Section 3.1.1. The error estimates then enter an error propagation scheme based on the thickness-flux relation of the shallow ice approximation (Equation 4), leading to

$$\delta H_i = \frac{1}{n+2}\left[-\frac{2}{n+2}B^{-1/n}(\rho g)^n \|\nabla h\|^n\right]^{-1/(n+2)}\|F\|^{-(n+1)/(n+2)}\|\delta F_i\| \quad i \in \{1,2\} \tag{12}$$

The final thickness error is the minimum of the two ($\delta H = \min(\delta H_1, \delta H_2)$). To calculate the error in flux ($\delta F$) we assume $\delta \dot{a} = 0.2$ m w.e. a$^{-1}$ and $\|\delta \boldsymbol{n}\| = 0.2$. Finally, to avoid unrealistic error estimates, we limit the uncertainty to 50% of the ice thickness in each pixel.

We calculate the error in the total ice volume as the product of the mean uncertainty in thickness and the total area of our domain.

**B.2 Error estimation of the second step**

We again follow Morlighem et al. (2014), assuming that the inaccurate flux field satisfies the mass conservation equation. However, in the second step the error propagates only through Equation 1. By analogy with section A.3.1, the error estimate from upstream and downstream is limited by

$$\nabla \cdot [(+\boldsymbol{u})\delta H_1] = \delta \dot{a} + \|\nabla \cdot [H\delta \boldsymbol{u}]\|$$
$$\nabla \cdot [(-\boldsymbol{u})\delta H_2] = \delta \dot{a} + \|\nabla \cdot [H\delta \boldsymbol{u}]\| \tag{13}$$

The considered input uncertainties are: apparent mass balance $\delta \dot{a} = 0.2$ m w.e. a$^{-1}$, flux direction $\|\delta \boldsymbol{n}\| = 0.2$, and velocity $\delta u = 20.0$ m a$^{-1}$. Finally, the thickness uncertainty $\delta H = \min(\|\delta H_1\|, \|\delta H_2\|)$. We again limit the uncertainty to 50% of the ice thickness in each pixel.

**C. Ice discharge calculation**

For the ice discharge calculation we used the same methods and the same set of flux gates as in Shahateet et al. (2023). The calculation of flux involves discretizing the flux gate into smaller evenly spaced sections, expressing the flux as a finite sum:

$$\phi = \int_S \rho \mathbf{u.dS} = \sum_i \rho L_i H_i f u_i \cos \alpha_i \tag{14}$$

where $\rho$ represents ice density, $L_i$ denotes the width of individual segments (set as $L_i = 200$ m), $H_i$ is the ice thickness of the segment $i$, $f$ is the ratio of surface to depth-averaged velocities, $u_i$ represents the speed of the ice on the individual segment $i$, and $\alpha$ is the angle between the velocity vector and the vector normal to the flux gate surface. The factor $f$ is assigned a value of 1.0, indicating that all motion is attributed to sliding rather than internal deformation. For further details on the ice-discharge calculation, refer to Shahateet et al. (2023).





## C.1 Error estimation of the ice discharge

Assuming that all contributing errors are independent and uncorrelated, we can estimate the statistically expected error using
error propagation from the individual error components as

$$\sigma_\phi = \sqrt{\sigma_{\phi_\rho}^2 + \sigma_{\phi_f}^2 + \sigma_{\phi_H}^2 + \sigma_{\phi_u}^2 + \sigma_{\phi_\alpha}^2} \tag{15}$$

where $\sigma_{\phi_\rho}$, $\sigma_{\phi_f}$, $\sigma_{\phi_H}$, $\sigma_{\phi_u}$, and $\sigma_{\phi_\alpha}$ are the ice discharge uncertainties associated to the individual terms of density, surface to depth-averaged velocity ratio, ice thickness, velocity modulus, and direction, respectively. As an example, in the case of the uncertainty associated to the ice thickness, it is calculated as

$$\sigma_{\phi_H} = \sqrt{\sum_i \left( \sigma_H \rho L_i f u_i \cos \alpha_i \right)^2} \tag{16}$$

## D. Additional figures





**Figure D1.** Close-up of the reconstructions of Carrivick, H&F, the one of the present work, Bedmap2, DeepBedMap, and Bedmachine v3 mostly on the Trinity Peninsula. Color of the ice-thickness scale has 60% transparency to enable the visualization of the optical image behind. We masked ice-thickness values lower than 10 m.





**Figure D2.** Normalized root-mean-squared deviation (NRMSD) of the six models evaluated in the present work (Carrivick, Huss and Farinotti, the present reconstruction, Bedmap2, DeepBedMap, and Bedmachine v3). See Shahateet et al. (2023) for a complete discussion on NRMSD.