# Peer review of "A reconstruction of the ice thickness of the Antarctic Peninsula Ice Sheet north of 70°S"

_EGUsphere, 2024_

## Referee Comment (RC1)

**Review of "A reconstruction of the ice thickness of the Antarctic Peninsula Ice Sheet north of 70$^{\text{o}}$ S" by Shahateet et al.**

August 30, 2024

**1 General**

Shahateet et al. set out to provide a new ice thickness map and associated volume estimate for the Antarctic Peninsula ice sheet North of 70°S. To that end, they rely on the method presented by Fürst et al. (2017) which in similar but slightly variable configurations has been applied in several previous studies (e.g. Fürst et al., 2018; Sommer et al., 2023; Fürst et al., 2024). They arrive at a volume estimate of $27.7\pm10\times10^3$ km$^3$ as well as an ice discharge total of $97.7\pm$ 15.4 km$^3$ based on their thicknesses and velocity observations.

The authors convincingly demonstrate that previous thickness inversion studies in the area have considerable shortcomings which warrants a new attempt. However, I remain somewhat sceptical as to whether this work really marks a significant improvement. This is because the chosen method has it´s specific weaknesses (c.f. major comments) and not the least because validation against the admittedly small set of available independent thickness observations is not very flattering (among others, a dramatic underestimation of high thicknesses is apparent). Notably, I feel that a thorough account of the limitations is not presented. With that said my concerns are detailed below which should be addressed before this manuscript should be considered further.

**2 Major comments**

**2.1 Account of limitations**

What I am missing throughout the manuscript is a more complete and honest account of limitations. The rather high errors after validation against the independent thickness observations and the dramatic underestimation of large thicknesses visible in Fig. 4 give good reason to be critical about the results presented here. And after downloading and looking at the thickness raster, I can spot several features in the thickness field that are not favorable performance indicators for a qualitative validation either (e.g. imprinted thickness observations, 'cracks' in the thickness field close to the southern domain boundary, outlet glaciers where the ice thickness is larger at the sides than in the center). These are likely issues with the method itself, and not related to the sometimes problematic quality of the assimilated thickness observations which the authors almost exclusively name as error source in the discussion and the abstract. I would like to ask the authors to be clear about the assumptions and associated limitations that come with using the SIA, with assuming down-slope glacier flow (if that is how the flow directions were determined, c.f. below), with the spatial interpolation of $B$, with the no-sliding condition where step two isn´t applied etc. (c.f. the following major comments). This necessitates at the very minimum adding a section to the discussion.

**2.2 The SIA inversion**

Section 3.1.1 describes what I call here the SIA inversion based on Fürst et al. (2017) where an ice flux is calculated and subsequently turned into an ice thickness via the SIA equation. In the description of the method, I miss what I believe to be a crucial part of this approach, namely of how to derive the flux directions. Eq. (2) relates the apparent mass balance to the flux *divergence*, whereas the following eq. (3) relies on the absolute volume flux. What is missing is how the flux directions are calculated

along which the mass is propagated down glacier. A flux direction error is mentioned in the appendix, so clearly flux directions are derived somehow, but this is not mentioned in the manuscript. In Fürst et al. (2017), the flux directions are down-slope after smoothing the DEM over a variable and thickness dependent length scale based on Kamb and Echelmeyer (1986). How is this done here?

Furthermore, I am not entirely convinced by the viscosity tuning and its spatial interpolation. Using the SIA at what I understood to be a 100 m resolution (this is not entirely clear from the manuscript, c.f. specifc comments) in a setting of such variable topography as the APIS means that the condition of large aspect ratio, mentioned by the authors in l.152f is often not met. As a result of that plus many other possible error sources, the ice viscosity parameter $B$ likely has little physical meaning at one particular point. Indeed, I could well imagine that at adjacent thickness observation locations along one measurement profile very different values for $B$ are needed to match the observations. Spatially interpolating these $B$ values between thickness observations, however, assumes that the area over which the interpolation is done has related errors to those at the observation locations. I am not convinced that this is the case. Not the least, I wonder whether attention is paid to flow catchment boundaries when the bilinear interpolation is done. What I mean here is that due to the variable topography of the APIS, two catchments with very different properties could be adjacent to each other and that if only one of them has thickness observations, interpolation of $B$ to the non-observed catchment is unlikely to be meaningful. (Or imagine a setting where a glacier cirque is adjacent to another glaciers main flow trunk, and only one of them has observations).

If you can demonstrate that the viscosity interpolation does yield a pattern that is likely to allow for a physically meaningful interpretation (e.g. by showing a map where this becomes evident), I strongly suggest to add this information to the manuscript to alleviate also other readers´ possible concerns.

Another question on the SIA inversion I have is on the boundary conditions at the 70°S boundary. If I understand correctly, you apply a no-flux condition here. Is that actually justified?

**2.3   Surface slope smoothing**

For both the SIA and the perfect-plasticity inversion, surface slope enters into the equation. It is an established fact that it is necessary to remedy the neglectance of membrane stresses when using these simplistic assumptions by smoothing the slope over a certain distance (e.g. Kamb and Echelmeyer, 1986). Indeed, a careful treatment of surface slopes is very important in the inversion context, as the slope is interpreted to be an expression of the ice flow over basal topography and nothing else. So, was smoothing or any other technique applied here? If so, please describe how (and discuss associated limitations).

**2.4   Perfect plasticity**

The same comment as for the $B$ interpolation. Is it really meaningful to spatially interpolate $\tau_d$? In this case, where $\tau_d$ should have a physical interpretation, wouldn´t it be much better to derive a relationship between between $\tau_d$ and certain local glacier characteristics that are likely to influence its distribution (e.g. surface slope, elevation) and then use that for interpolation? You are rather hard on Carrivick et al. (2018) for relying on the relationship by Dridger and Kennard (1986), but I am not convinced that your approach is much superior.

Again, a map (possibly in the appendix) of the interpolated $\tau_d$ would allow the readers to judge themselves.

**2.5   Second step**

Where you deem the velocities to be reliable, you use them to apply the mass conservation equation directly, thus circumventing the need to model ice dynamics. This is an appealing approach, but I think some clarification is needed. First, how did you determine the threshold of 200 m yr$^{-1}$ at which you classify the velocities as reliable (notably, this is twice as much as in, e.g., Fürst et al. (2024))? Second, your SIA and perfect-plasticiy approaches assume no sliding, while you assume 100% sliding

in those areas where you apply the second step. This obviously results in a huge discrepancy at the boundaries. This needs to be mentioned, alongside an explanation of how you think this influences the result. I might think that minimization equation (6) helps you with creating a smooth transition. However, that makes me wonder about your statement in l.188 where you mention that inconsistencies with the mass balance are the reason why eq. (6) is needed. Isn´t it also (and to a large part) the inconsistency in assumed ice flow physics that are remedied by eq. (6)? Finally, how do you justify the assumption of no sliding up to a velocity of 200 m yr$^{-1}$?

**2.6   Input data**

I was surprised to see that you purposefully choose to create a larger temporal mismatch in your input data than what appears necessary. It seems to me as if you could have focused on the period 2013-2017 for surface elevation change, SMB and velocities. This would also have aligned well with the DEM (2012-2020 according to l.316; needs to be mentioned in sec.2.2) and the thickness observations (2009-2021). Instead you extended the SMB timing to 2011-2020 and shortened the velocity time series to 2015-2017. Why is that?

In addition, you resample the thickness observations to 500 m, although you model on a 100 m grid. You mention that you did tests to arrive at your decision(l.105), but it is unclear what tests. So, is there any clear reason to not use thickness observations at a 100 m posting?

And speaking of resolution, your results certainly do not reflect horizontal variations in ice thickness of 100 m. This isn´t physically possible as horizontal features smaller than around one ice thickness don´t leave any surface expressions (Gudmundsson, 2003). Please discuss this limitation in the manuscript.

Lastly, you should be more clear on how you treat ice shelves. From Fig.1 I take that you didn´t model Larsen C, but how did you deal with other ice shelves? If they were not modelled, how did you derive the grounding line location? If I am not overlooking something this information is currently not very clear.

**3   Minor comments**

Is there a specific reason why the authors do not choose to provide a raster with bed elevations? Many potential users will need these instead of thicknesses (e.g. for ice flow modelling), and would need to find and download the exact same DEM as used in this study if they wanted to derive them themselves.

Almost no word is lost on the derivation of the error map save for in the appendix. As the method assimilates almost all thickness observations available in the area it is even more important that one can trust the error calculations when judging the quality of this product. Consequently, I suggest adding a plot with the error map, preferably with inset maps zooming in on specific areas to allow readers to get an impression of where largest errors can be expected locally. Related to that, could you comment on your choice of limiting uncertainties to max. 50% of the ice thickness (l.617 ); Fig. 4 shows that this isn´t necessarily justified.

**4   Specific comments**

L.3: "Such *an* approach..."

L.9: as mentioned above, be a bit more critical about the limitations of your work

L.26: the acronym GMB is only used once and can be removed

L.35: Consider to cite the two studies you refer to to allow the reader to recognize which ones you are talking about

L.35f.: thickness maps aren´t physically based methods (they can be a product of them); rephrase

L.37: Some inversion studies (e.g. Millan et al., 2022) have used the SIA in combination with a "sliding law" (i.e. an assumed sliding contribution to flow speeds). In that case, thickness values aren´t automatically inversely proportional to slope anymore. Please mention such approaches. Also consider referring to studies that simultaneously infer the basal friction field (Jouvet, 2023; Frank et al., 2023, e.g.) somewhere in the introduction as these also are techniques that circumvent the problem described here.

L.51: I don´t find it justified to claim that a study area such as the APIS is "the most challenging due to its complex geometry". While it is true that input datasets often are adversely impacted by complex topography, it is usually much easier to recover the subglacial bed in areas with pronounced topography as opposed to in the interior of flat ice caps or ice sheets. This has to do with how basal perturbations are transmitted to the surface (e.g. Gudmundsson, 2003).

L.61: ice thickness *observations*

L.75: Somewhere in this paragraph mention the acquisition date of the DEM (instead of in l.361)

L.79: For the reader, it is not directly clear that 100 m is not only the resolution of the input DEM, but indeed of your entire modelling. Mention this somewhere explicitly.

L.116: I suggest to also reference previous studies that have relied on the Fürst et al. (2017) approach (c.f. above)

L168f.: Perhaps a matter of taste, but typically people call the $\tau$ in the perfect plasticity equation basal shear stress or yield stress (e.g. Linsbauer et al., 2012). Your formulation isn´t wrong, but I find the wording a bit strange. In l.275 you actually call it basal shear stress yourself.

L.188: You name mass balance inconsistencies as the reason to apply eq. (6), but $u$ is also a control variable. And c.f. comment above

L.197: What is the difference between "scaling down the velocity derivative by a factor of 0.05" and simply multiplying your choice of $\gamma_U$ with 0.05?

L.205: "..according to the assumption, SIA, or PP respectively." Rephrase

L.217: You claim that the discussion will focus on the combined result, but actually you make many more references to the individual results later on

L.224f: From the caption of Fig.3 I take that this is after applying step 2. If so, it does not show what it is claimed to show in the text here. Or is the caption of Fig.3 incorrect? Regardless, the reader doesn´t know where the fast-flowing regions are (c.f. comment on figures), so that figure doesn´t prove the point the authors are making here

L.278: It is a well-established fact that larger glaciers generally have a larger mean ice thickness. I don´t think this is something to be too critical about.

L.301: I believe that Figure 6 is mentioned before Fig. 4 and 5. Then usually it should also be shown first. Furthermore, since Figure 6 doesn´t show any thickness observations (and hence independent validation), I don´t think it can be used to make the point that certain models likely underestimate ice thicknesses.

L.336: I do find it a bit too euphemistic to call a NMAE of around 40% a "relatively good performance". It is a rather worrisome sign and should initiate a thorough discussion on the limitations of this study beyond thickness observation uncertainty.

L.346ff: I don´t 100% follow on how you derive ice thickness "observations" from the bathymetric data. Specifically, how do you deal with those areas that you identify as floating (L.347)? My impression from your manuscript was that you don´t model ice shelves (c.f. above), so you shouldn´t have any thickness values there. Or do you and compare your "grounded" thicknesses against the "floating" thickness observations? Please clarify!

L.361: as mentioned above, give the date of the REMA DEM somewhere in the methods section

L.375: thick –> thickness

L.386: remove "all" since you are referring to only three of the five results that you are discussing

L.387: you appear to take it as a prove of quality that three methods agree on thick ice over the plateau. However, the methodology of these approaches all have particular weaknesses on flat plateaus (e.g. due to low surface slopes), and you don´t mention any thickness observations that really would prove that thick ice can be found here. I thus suggest to write a bit more carefully

L.417: be more specific that you refer to NMAE, instead of only writing "errors"

L.423: In the conclusions you offer new speculations on what could be wrong with the thickness observations. This should be mentioned earlier.

Appendix A.: This section appears rather disconnected from the rest of the text. When you talk of "THE" zero transitions, one would expect that they were mentioned earlier. However, that isn´t the case I believe. So please provide a bit more context here.

**5    Figures and Tables**

In general, I would encourage the authors to consider providing a figure where all input data is shown. At least, though, I would strongly recommend to add a figure where those areas that are treated with step two are indicated since this is something the authors refer to a lot.

Fig.1: Distinguish more clearly between land and ice shelf.

Fig.2: I appreciate this nice overview. But I believe that it should read "..the PP approach uses the glacier outlines and elevation field.." instead of "elevation change field" in the caption.

Fig.3: I suggest to write explicitly what was subtracted from what for the difference plot. Furthermore, I don´t find the 60% transparency helpful in any way (here and in other figures). Most of the time, as in this case, the background optical image isn´t visible anyways.

Fig.4: Title of panel a) should read "Flask and Leppard glacier" I believe; also, please add the 1:1 line to allow for a direct interpretation without having to read the labels and increase the font size of all labels. The x-axis labels should indicate that observations are shown, not just the name of the study

Fig.5: I think panel c) (zoomed in on the upper part) is sufficient to make the point that OIB thicknesses can be unreliable, no need to present such a large figure

Fig.6: Consider using a color scale that is more clearly distinguishable from the background optical image. Also, if this is meant to demonstrate that some products are more reliable than others thickness observations should be shown as independent verification.

Table 2: I may miss something here, but are the values really what was calculated with eq. (7)? If so, why is the value for this study not shown? Or do these numbers represent some ratio relative to

this study? If so, please clarify or change eq. (7) accordingly.

**References**

Frank T, van Pelt WJJ, Kohler J. 2023. Reconciling ice dynamics and bed topography with a versatile and fast ice thickness inversion. The Cryosphere 17:4021–4045. doi:10.5194/tc-17-4021-2023.

Fürst JJ, Farías-Barahona D, Blindow N, Casassa G, Gacitúa G, Koppes M, Lodolo E, Millan R, Minowa M, Mouginot J, Ptlicki M, Rignot E, Rivera A, Skvarca P, Stuefer M, Sugiyama S, Uribe J, Zamora R, Braun MH, Gillet-Chaulet F, Malz P, Meier WJH, Schaefer M. 2024. The foundations of the Patagonian icefields. Communications Earth & Environment 5:1–10. doi:10.1038/s43247-023-01193-7.

Fürst JJ, Gillet-Chaulet F, Benham TJ, Dowdeswell JA, Grabiec M, Navarro F, Pettersson R, Moholdt G, Nuth C, Sass B, Aas K, Fettweis X, Lang C, Seehaus T, Braun M. 2017. Application of a two-step approach for mapping ice thickness to various glacier types on Svalbard. The Cryosphere 11:2003–2032. doi:10.5194/tc-11-2003-2017.

Fürst JJ, Navarro F, Gillet-Chaulet F, Huss M, Moholdt G, Fettweis X, Lang C, Seehaus T, Ai S, Benham TJ, Benn DI, Björnsson H, Dowdeswell JA, Grabiec M, Kohler J, Lavrentiev I, Lindbäck K, Melvold K, Pettersson R, Rippin D, Saintenoy A, Sánchez-Gámez P, Schuler TV, Sevestre H, Vasilenko E, Braun MH. 2018. The Ice-Free Topography of Svalbard. Geophysical Research Letters 45:11,760–11,769. doi:10.1029/2018GL079734.

Gudmundsson GH. 2003. Transmission of basal variability to a glacier surface. Journal of Geophysical Research: Solid Earth 108. doi:10.1029/2002JB002107.

Jouvet G. 2023. Inversion of a Stokes glacier flow model emulated by deep learning. Journal of Glaciology 69:13–26. doi:10.1017/jog.2022.41.

Kamb B, Echelmeyer KA. 1986. Stress-Gradient Coupling in Glacier Flow: I. Longitudinal Averaging of the Influence of Ice Thickness and Surface Slope. Journal of Glaciology 32:267–284. doi:10.3189/S0022143000015604.

Linsbauer A, Paul F, Haeberli W. 2012. Modeling glacier thickness distribution and bed topography over entire mountain ranges with GlabTop: Application of a fast and robust approach. Journal of Geophysical Research: Earth Surface 117. doi:10.1029/2011JF002313.

Millan R, Mouginot J, Rabatel A, Morlighem M. 2022. Ice velocity and thickness of the world's glaciers. Nature Geoscience 15:124–129. doi:10.1038/s41561-021-00885-z.

Sommer C, Fürst JJ, Huss M, Braun MH. 2023. Constraining regional glacier reconstructions using past ice thickness of deglaciating areas – a case study in the European Alps. The Cryosphere 17:2285–2303. doi:10.5194/tc-17-2285-2023.

---

## Author Response (AR1)

Dear Dr. Barrand,

We sincerely appreciate the time and effort you and the anonymous reviewers have dedicated to evaluating our manuscript. Please find attached the final version of the manuscript, along with a version highlighting the changes in blue.

Best regards,

Kaian Shahateet (on behalf of all authors)